# The Impact of Minimal Intervention Dentistry on Patient-Reported and Observation-Based Outcomes in the Pediatric Population: A Systematic Review and Meta-Analysis

**DOI:** 10.3390/healthcare11162241

**Published:** 2023-08-09

**Authors:** Hilton Hiu Chun Chiu, Phoebe Pui Ying Lam, Cynthia Kar Yung Yiu

**Affiliations:** Paediatric Dentistry, Faculty of Dentistry, The University of Hong Kong, Pokfulam, Hong Kong SAR, China; hhcchiu@connect.hku.hk (H.H.C.C.); pyphoebe@hku.hk (P.P.Y.L.)

**Keywords:** minimal intervention dentistry, patient-reported outcomes, dental anxiety, preschool children, systematic review, meta-analysis

## Abstract

This review aimed to systematically investigate the effect of minimal intervention dentistry on patient-reported and observation-based outcomes of anxiety, pain and patient cooperation in the pediatric population. Microinvasive treatments (MITs) were compared to conventional treatments, home-based and professionally applied non-invasive treatments (NITs), and between MITs. Two reviewers independently screened studies from four electronic databases, extracted data, assessed the risk of bias and certainty of evidence with the GRADE approach. Ultimately, 26 studies were included for qualitative synthesis, with the data from 12 studies being incorporated into the meta-analysis. No statistically significantly differences in terms of anxiety were noted between MITs and conventional treatments, or between MITs from the meta-analyses. The treatment durations of MITs were not necessarily shorter than conventional treatments but varied with the use of local anesthesia, behavioral and clinical approach, and other confounding factors. The certainties of evidence were deemed low due to high risk of bias of the included studies. NITs evoked less anxiety and pain compared to MITs. Minimal intervention dentistry is an alternative treatment to manage dental caries among children and does not arouse different levels of dental anxiety and pain compared to other treatment modalities. However, further well-designed studies are required to draw an evidence-based conclusion.

## 1. Introduction

Dental caries is recognized as one of the most prevalent chronic diseases that affect the population [1] and its treatment will involve a dental visit. Yet, one of the greatest obstacles in providing appropriate treatment for pediatric patients is dental anxiety, which is a general state of apprehension in preparation for a negative experience [2]. Unfortunately, such anxiety may impede the provision of treatment due to lack of compliance, lower utilization of dental services [3], and ultimately, poorer oral health status [4]. Moreover, if the patient could not cooperate in an outpatient setting, anesthesia with hospital care is indicated which incurs higher costs and greater risks for the patient and their caretakers [5]. Hence, dental anxiety is a fundamental issue that clinicians must tackle when treating the pediatric population.

The conventional methods of restoring dental caries include various aspects which may trigger distress in patients. For example, local anesthesia application, rubber dam isolation, the use of electric motor drills and rotary burs are all elements that may be included during restoration of carious teeth. Previous studies have demonstrated that discomfort associated with injections, the use of dental handpiece and sensitivity experienced during carious tissue removal may cause trepidation and fear in pediatric patients [6]. Hence, when providing treatment to manage dental caries, removing the above stimuli is essential for lowering the levels of anxiety experienced by children.

In considering the difficulties mentioned above, minimal intervention dentistry has recently become a novel component of the caries management protocol. According to the FDI 2016 policy statement, the concept of minimal intervention dentistry is to “conserve re-mineralizable and intact tooth tissue to help retain teeth throughout life” [7]. The effectiveness of minimal intervention dentistry to prevent the development of caries and minimize dental anxiety in the pediatric population has been widely reported in the literature. A recent Cochrane review has suggested that fissure sealants on permanent molars may reduce caries prevalence by 11–51% [8]. Atraumatic restorative technique (ART) is also a principle of minimal intervention dentistry introduced 30 years ago to restore cavitated carious lesions in areas without rotary instruments, electricity and water [9]. Hand excavators are used to remove soft and demineralized tissue; subsequently, the cavity is cleaned and restored with an adhesive dental material using finger pressure [10]. Previous studies have reported that ART may reduce pain experience and dental anxiety compared to conventional restorations (CRs) using the same material [10,11]. Techniques such as stainless steel crowns placed using the Hall Technique (HT) which requires no local anesthesia, and no preparation was introduced in the mid-1990s as a method to seal and arrest caries [12]. The literature has reported that crowns fitted using HT may reduce patient’s discomfort compared to CR [13] and randomized controlled trials have shown high success rates at over 90% [14,15]. Given the efficacy of minimal intervention dentistry approach and the potential to alleviate dental fear and reduce discomfort, it should be considered as an alternative to manage dental caries in anxious pediatric patients.

There have been numerous studies that explore the clinical efficacy of minimal intervention dentistry on caries progression and discomfort. However, there have not been many comprehensive reviews that compared the effect of minimal intervention dentistry to conventional restorative techniques on multiple patient-reported outcomes and observation-based outcomes (OBOs) such as anxiety, discomfort, cooperation, time and adverse effects. Patient-reported and observation-based outcomes are equally important to facilitate patient-centered care.

Therefore, a systematic review was carried out with the aim to compare the patient-reported and observation-based outcomes of minimal intervention dentistry with other treatments, hoping to answer the following question: in the pediatric patient population, will minimal intervention dentistry approaches to manage dental caries have an impact on dental anxiety, discomfort, patient cooperation and other patient-reported and observation-based outcomes, as compared to no treatment, conventional restorative techniques and between micro-invasive treatments (MITs)?

## 2. Materials and Methods

The systematic review was conducted following the guidelines outlined in the Cochrane Handbook for Systematic Reviews of Interventions and Preferred Reporting Items for Systematic Reviews [16] and Meta-Analyses (PRISMA) statement [17]. The review was registered in PROSPERO (Registration ID: CRD42021227838). The following PICOS statement was recommended.

### 2.1. Type of Participants (P)

Studies which recruited preschool and primary school children under the age of 12 with at least one cavitated or non-cavitated caries lesions without any signs of pulpal or periapical involvement were included. Children with systemic medical conditions or taking long-term medications or requiring special needs attention were excluded.

### 2.2. Type of Intervention (I) and Control/Comparison (C)

This review included studies where minimal intervention dentistry techniques were employed to manage dental caries. Ultimately, these operative procedures aim to stop the progression of caries by conserving re-mineralizable tissues. The types of interventions include micro-invasive treatments (MITs) that involve the application of an acid to prepare or condition the tooth surface, followed by sealing the decayed surface on top [18]. The demineralized decay is either left in place or removed with hand instruments only [18]. The intervention is micro-invasive as at least a few micrometers of tooth structure will be removed during the procedure [18]. The interventions include and are not limited to resin infiltration, fissure sealants (resin-based or glass ionomer-based), ART restorations with the use of hand excavation, restorations completed using step-wise or selective caries removal technique and preformed crowns placed with HT.

This review included four comparisons. The first comparison is comparing MITs with conventional operative treatment or “treatment as usual” to manage dental caries. Such methods include using micromotor with handpieces and burs to remove carious tissue before restoring the cavity with amalgam or other adhesive restorative materials. Local anesthesia (LA) and rubber dam (RD) may be needed to facilitate the restorative process. However, pulp treatment was excluded from the control group as treatment time will be lengthened significantly. The second comparison were between two MITs. The last two comparisons were between MITs with home-based non-invasive treatments (NITs), and professionally applied NITs. NITs promote remineralization without restoring the lesion or cavity [18], and no decayed tissue are removed during the process to manage the caries lesion and control the mineral balance [18]. Home-based NITs include oral hygiene instructions or dietary advice [18], whereas professionally applied NITs include and are not limited to professionally applied fluoride and other topical agents such as sodium fluoride varnish, stannous fluoride gel, acidulated phosphate fluoride (APF), and silver diamine fluoride (SDF).

### 2.3. Types of Outcome Measures (O)

The primary outcomes evaluated were the patient-based or observation-based outcomes that assess the patient’s level of cooperation, dental anxiety and pain. The outcomes could be measured using four main channels: self-report assessment, parental proxy assessment, observation-based assessment and physiological assessment [19]. For this review, scales using self-report assessment and observation-based assessment were included as the patients-based methods. Examples of the scales include, and are not limited to, Frankl Scale (FS), Venham Scale (VS), Facial Image Scale (FIS), Wong Baker Facial Scale (WBFS), Visual Analogue Pain Scale, etc. Studies measuring the peri-operative or post-operative score after delivering the treatment were included in the review. The secondary outcomes evaluated were the time and presence of adverse outcomes.

### 2.4. Types of Studies (S)

Parallel or split mouth randomized controlled trials and controlled clinical trials of any duration were included in this review.

### 2.5. Information Sources and Literature Search

The initial search was conducted systematically using the following four electronic databases (Cochrane Central Register of Controlled Trials (CENTRAL); Ovid Embase; PUBMED; Web of Science), using the broad keywords and specific medical domains from inception up to 31 August 2022 (Appendix A). Only publications with full English text were included in the review. The reviewers also conducted hand searches by assessing references lists from relevant systematic reviews in the Cochrane Library (Cochrane Database of Systematic Reviews) and other past reviews to confirm that no relevant studies were overlooked.

### 2.6. Study Selection

Two reviewers (H.H.C.C. and P.P.Y.L.) assessed the articles retrieved from the databases independently based on the titles, keywords and abstracts. Subsequently, the potential eligibility was assessed separately. Agreements between the reviewers were determined using Cohen’s kappa coefficient (k). A third reviewer (C.K.Y.Y.) was consulted for assessing the final eligibility if there were disagreements between the two authors.

### 2.7. Data Collection and Measurement of Primary Outcomes

Data were extracted independently by the two reviewers (H.H.C.C. and P.P.Y.L.) using standardized data extraction forms. Data including study characteristics (trial design, year and duration), participants (location, number, age, gender, inclusion and exclusion criteria), intervention (number in the group and type of treatment), primary outcomes (score of patient anxiety/cooperation scale) and secondary outcomes (time and presence of adverse outcomes) were recorded in the input forms. When evaluating the primary outcomes, the categorical data score of the scales of anxiety and cooperation were extracted in both the intervention and control groups. When available, the mean score and standard deviation were extracted or calculated from the report and included in the table for meta-analyses.

### 2.8. Risk of Bias in Individual Trials

Risk of bias for each study were evaluated using the revised Cochrane risk of bias tool for randomized trials (RoB 2.0) [20]. The tool included five domains of bias with prompting questions to allow the reviewer to assess each study’s level of bias. The five domains include (I) bias arising from the process of randomization, (II) bias due to deviation from the intended interventions, (III) bias due to missing outcome data, (IV) bias in the measurement of outcome and (V) bias in the selection of reported result.

### 2.9. Data Synthesis and Analyses

Studies that compared the level of anxiety, pain or cooperation of patients under treatment using the scales mentioned above were included for data synthesis and analysis. The standardized mean difference (SMD) was used as a summary statistic as the current review is investigating the outcomes based on various continuous psychometric scales. When the mean scores were not available for data extraction, other statistical data were extracted for comparison, such as distribution. Meta-analyses would be conducted with Stata version 13.1 if sufficient data are available. If less than five studies were included, the fixed effects model was employed, and random effect was used if more than five studies were included [21]. Sensitivity analysis was conducted if more than five studies were included.

### 2.10. Assessment of Heterogeneity

The assessment of heterogeneity was performed according to the guidelines stated in the Cochrane Handbook of Systematic Reviews of Intervention [22]. The level of heterogeneity between the literature was measured using I^2^ statistic. The level of significance for statistical heterogeneity was determined at *p* < 0.05 and it was calculated using a Chi-square test. Heterogeneity was determined as significant when I^2^ > 60% and *p*< 0.05 [22].

### 2.11. Assessment of Publication Bias

The assessment of publication bias was in line with the Cochrane Handbook of Systematic Reviews of Intervention [23]. The assessment would use funnel plots when there are more than 10 studies included in a particular outcome, but funnel plots and publication bias were not carried out as there was no meta-analysis involving more than 10 included studies.

### 2.12. Assessment of Evidence 

Two reviewers (H.H.C.C and P.P.Y.L.) independently assessed the certainty and quality of evidence for the outcomes using the Grading of Recommendations Assessment Development and Evaluation (GRADE) approach [24]. The studies were subject to downgrading when the data presented with serious risk of bias, imprecision, inconsistency, indirectness and publication bias. 

## 3. Results

This section is divided into subheadings. It should provide a concise and precise description of the experimental results, their interpretation, as well as the experimental conclusions that can be drawn.

### 3.1. Study Selection

A systematic literature search found 1101 studies and after removing the duplicates, 305 remained for screening. After screening for titles and abstracts, 43 were retrieved for full text reading (Kappa k: 0.986). Out of the 56 texts, 30 were excluded due to the following reasons: irrelevance (*n* = 2), non-randomized controlled trial (*n* = 2), no retrieval of English full text (*n* = 7), registered protocol only (*n* = 1), reporting oral health-related quality of life (*n* = 8), clinical success (*n* = 5), subjects were adolescents (1) or with special healthcare needs (1) and caregiver-reported outcomes (*n* = 3). Subsequently, only 26 [14,25,26,27,28,29,30,31,32,33,34,35,36,37,38,39,40,41,42,43,44,45,46,47,48,49] studies were included for the final qualitative and quantitative analysis and data extraction (Kappa k: 1) (Appendix A). A summary of the PRISMA flowchart [50] for identification and screening the eligibility of the papers, final inclusion of the studies and reasons for exclusion are illustrated in Figure 1.

### 3.2. Study Characteristics 

The characteristics of the included studies are illustrated and summarized in Table 1. Ultimately, the 26 included studies [14,25,26,27,28,29,30,31,32,33,34,35,36,37,38,39,40,41,42,43,44,45,46,47,48,49] represented 4699 participants from 5 Asian countries [34,37,38,43,45], 1 Australasian country [29], 8 European countries [14,27,31,35,42,46,47,48], 11 Southern American countries [25,26,28,30,32,33,39,40,41,44,49] and 1 African country [36]. Six of the studies were published before 2010 [14,35,43,46,47,48] and the remaining fifteen studies were published between 2011 and 2022 [25,26,27,28,29,30,31,32,33,34,36,37,38,39,40,41,42,44,45,49]. Almost all the studies conducted their treatment in a dental clinic or university dental clinic, while only two studies [31,41] conducted the treatment in schools or a mobile outreach dental unit. The age of the participants in the included studies ranged from 3 to 10 years. 

Ten [25,27,29,30,31,32,36,42,44,49] of the studies used Facial Image Scale (FIS) or variants of this scale (e.g., Delighted Terrible Faces scale, Visual Analogue scale) to measure the participant’s level of anxiety. An additional seven [26,28,33,38,40,42,44] studies employed the WBFS or the Hedonic scale to assess the child’s pain, discomfort and acceptance during the treatment. Eight studies [37,41,42,43,45,46,47,48] utilized FS, VS or Venham Picture Test (VPT) to evaluate the child’s cooperation. Of the twelve studies included in the meta-analysis [14,25,28,29,34,35,36,38,39,44,46,48], two studies presented their findings in categorical data, while the rest presented their findings in continuous data, which to be extracted for performing the quantitative syntheses. 

The MITs of the included studies comprised ART [9], fissure sealants, resin infiltration, and stainless steel crowns placed using HT [14]. Such interventions are in line with the principles of remineralization and repair outlined in the FDI policy statement [7] on MID. Seventeen [25,26,28,29,30,34,35,37,38,39,41,43,44,46,47,48,49] studies used ART restorations as their intervention, and of these studies, ten studies compared ART with CR [25,26,29,35,39,43,44,46,47,48], three compared ART with cavity modifications [26,37], two used ART together with oral health education [41] or Brix3000™ papain gel [33] and compared to ART alone, and three [28,34,38] compared ART restorations to HT in terms of patient-reported outcomes, observation-based outcomes, and anxiety using the abovementioned quantitative scales. Another two [27,40] studies assessed resin infiltration as their intervention. The remaining four [31,32,36,42] studies used fissure sealant and HT as the intervention.

In summary, sixteen studies compared MITs to conventional restorative treatment [14,25,26,29,32,34,35,36,39,42,43,44,45,46,47,48], six studies focused on comparisons between different MITs [25,26,28,37,38,42] and four studies compared MITs to either home-based or professionally applied NITs [27,30,40,49].

### 3.3. Risks of Bias of Included Studies 

The risk of bias of the included studies are presented in Figure 2. A comprehensive evaluation of the risk of bias could not be executed due to incomplete reporting and lack of data in the included studies. The revised Cochrane risk of bias assessment tool [20] was used and all the studies were found to have a high risk or with concerns in terms of overall bias. This is likely because blinding could not be achieved as either the outcome assessors or the participants were fully aware of the treatment that was performed. This was especially of concern when studies only applied observer-rated scale, for instance, FS and VS, and the operator or dental assistant had to personally assess the child’s cooperation while completing the procedure; hence, blinding is unmanageable. 

### 3.4. Comparison of MITs to Conventional Restorative Treatment

#### 3.4.1. ART vs. CR

Summary of the results of primary outcome is shown in Appendix A. Among the ten studies [25,26,29,35,39,43,44,46,47,48] that compared ART with CR, two studies [29,48] categorized and compared the proportions of anxious children in ART and CR with LA groups, respectively (Figure 3). Even though the two studies utilized different scales to rate the anxiety level of children [29,48], both identified no significant difference between the proportion of children being anxious when receiving ART and CR (RR, 0.88; 95% CI, 0.74, 1.05; *p* = 0.149; I^2^, 66.5%; *p* = 0.084). However, certainty of evidence was very low due to high risk of bias of the included studies, inconsistency and indirectness (Table 2).

Two studies [25,46] evaluated the mean difference of anxiety levels in two different self-rated scales between ART and CR; both studies consistently identified no significant difference in the mean anxiety and pain scores between the two groups (SMD, 0.01; 95% CI, −0.20, 0.21; *p*= 0.959; I^2^, 0.0%; *p* = 0.824) [25,46] (Figure 4). However, the two studies were with high risk of bias, with one performing the two interventions at different settings [46], while the other did not standardize the use of LA in the control group [25]. The certainty of evidence was deemed low due to high risk of bias of the included studies and imprecision (Table 2).

Two studies [26,44] assessed the pain level of children receiving ART and CR with WBFR. Tavares et al. (2018) [44] compared the median WBFR score and reported that children in the CR group had more pain when no LA was given. Abreu et al. (2011) [26] reported no difference in WBFR between ART and CR, but significantly more children in CR group requested LA and LA was given upon request in this study (Table 2) [26].

Three studies [35,39,44] compared the mean time required for ART and CR; however, the results were very heterogenous as the methods of how the CR were conducted were totally different (I^2^, 97.2%; *p* < 0.001.) Luz et al. (2012) [39] performed the CR with LA and RD, whereas Eden et al. (2006) [35] and Tavares et al. (2017) [44] conducted it with rotary instruments only. Hence, subgroup analyses comparing ART with CR performed without LA and RD identified that CR required significantly shorter treatment time than ART (WMD, −1.82; 95% CI, 1.35, 2.28; I^2^, 0.0%; *p* = 0.361). The quality of evidence was rated as moderate as results were generated from randomized controlled trials, even though one of the included studies had a high risk of bias (Figure 5, Table 2).

Two studies [43,47] measured and compared the pulse rate of subjects receiving the two interventions. Although data cannot be pooled due to difference in reporting, both studies reported no significant difference in the pulse rate for children receiving ART or CR at each treatment phase [43,47].

#### 3.4.2. HT vs. Conventional Stainless Steel Crowns (CSSC)

Three studies [14,34,36] were included for comparing the patient-reported outcomes and observer-based outcomes between HT and CSSC (Appendix A). Children receiving HT had a significantly lower mean FIS score, but their WBFS were similar compared with those receiving CSSC (Appendix A). Two studies [14,34] investigated the time required for placing HT and CSSC on a total of 338 teeth but found heterogenous results (I^2^ 97.9%; *p* < 0.001) (Appendix A). Ebrahimi et al. (2020) [34] just calculated the crown preparation and insertion time and reported that HT required significantly shorter time (mean difference (MD), −8.9 min; 95% CI, −11.1 min, −6.72 min). Meanwhile, Innes et al. (2007) [14] included the time for separator placement and patient instructions in the calculations, and reported no significant difference between the two groups (MD, −0.90; 95% CI, −0.82, 2.62).

#### 3.4.3. HT vs. CR

Two studies [42,45] compared the patient-reported outcomes and observer-based outcomes between HT and CR, and both reported that children in the HT group had better behavior measured with FBRS. However, data cannot be pooled due to missing reporting data in one study.

#### 3.4.4. Fissure Sealant vs. CR

One included study [32] investigated and found no difference in FIS scores between the two treatment groups, fissure sealant and composite CR (*p* = 0.650). However, a full set of data was not available as dental anxiety was a secondary outcome measure in this clinical trial.

### 3.5. Comparison between MITs

#### 3.5.1. ART vs. Cavity Modification (CM)

Three studies (25, 26, 37) compared the discomfort and anxiety of children when receiving ART versus CM, but meta-analysis could not be performed due to heterogeneity of the reporting methods. No significance difference was found between the two interventions in terms of self-rated mean FIS scores [25] and WBFR [26]. However, the observer-stated that the overall mean Venham scores of the ART group was significantly lower than the CM group [37].

#### 3.5.2. HT vs. ART

Meta-analysis was performed on three included studies [28,34,38] comparing the WBFR of children immediately after receiving ART and HT (Figure 6). The findings from the meta-analysis identified no significant difference in pain level immediately after post-operative (SMD, −0.24; 95% CI −0.50, 0.01; *p* = 0.061). For peri-operative pain level, only one study reported that children felt more pain during the treatment procedures of ART [38], especially when caries were being extracted from the cavities with manual instruments (*p* < 0.001).

All included studies [14,25,26,27,28,29,30,32,33,34,35,36,37,38,39,41,42,43,44,45,46,47,48,49] did not compare and report the treatment time for HT and ART. Regarding the associated adverse effect, HT resulted in an immediate increase in occlusal vertical dimension (OVD) between 0.63 ± 0.61 mm and 1.45 ± 0.87 mm [28,38]. However, the increased OVD was settled after four weeks [28], and both studies did not report any discomfort associated with the increased OVD from subjects receiving HT [28,38]. No other associated side effect has been reported, except that the parents of children receiving HT showed more aesthetic dissatisfactions (24%) than those of children receiving ART (5%) (28).

### 3.6. Comparisons of MITs to NIT

#### 3.6.1. ART vs. SDF

Two studies [30,49] compared the dental anxiety of children between ART and SDF with FIS; both studies identified no significant difference in dental anxiety level immediately postop, but meta-analysis could not be performed due to differences in reporting. Vollu et al. (2019) recorded and compared the time required [49], and reported that SDF application required significantly shorter mean treatment time (6.97 ± 1.31 min) than ART (13.88 ± 4.25 min; *p* < 0.001) [49].

#### 3.6.2. MITs vs. Professionally Applied NITs

One randomized controlled trial [31] conducted by the National Health Service investigated the acceptability of fissure sealants and sodium fluoride varnish (NaFV) for 6 to 7-year-old children. Chestnutt et al. (2017) used the Delighted Terrible Faces Scale immediately after the procedure to identify the patient’s level of acceptability and found a statistically significant difference [31]. Children receiving fluoride varnish were more likely to be identified as being happy compared to those receiving fissure sealants (OR = 0.38; 95% CI 0.29–0.50; *p* = 0.001) [31]. This notion is further supported by Mattos-Silveira et al. (2014) who found that children in the resin infiltration group displayed higher levels of discomfort compared to the SDF group (Adjusted RR= 0.29; 95% CI 0.12–0.71; *p* = 0.006) [40]. The above two studies identified that children who underwent professionally applied treatment such as fluoride application felt less discomfort compared to MIT. The current evidence suggests that NITs will render less or no difference in anxiety and discomfort compared to microinvasive treatment methods. No included studies reported any other adverse effect associated with MITs and NIT.

#### 3.6.3. MITs vs. Home-Based NITs

Only two studies [27,40] investigated patient-reported outcomes, observer-based outcomes and dental anxiety for these two types of interventions. Ammari et al. (2017) examined the patient-reported dental anxiety for patients receiving resin infiltration and only flossing and toothbrushing using a split-mouth design [27]. Low anxiety levels were identified for patients receiving resin infiltration both before and after the treatment, with 84% of participants scoring 1 or 2 (positive faces) and only 4% scoring 3–5 (negative faces) with FIS [27]. However, data regarding dental anxiety from the control group of only flossing and toothbrushing could not be collected as it was performed by the parents at home. Hence, factors such as a different environment and parents conducting the procedure, will have affected the outcome and the participant’s anxiety. No comparison could be carried out in the study; thus, no definite conclusion could be drawn.

Another randomized controlled trial [40] also examined the differences between resin infiltration and just flossing and OHI, and found statistically significant higher levels of pain in participants receiving resin infiltration than those in the control group through WBFS. The mean treatment time required for resin infiltration was 11.29 ± 1.16 min [27]. Through the Poisson regression analyses, children who only received flossing and OHI were less likely to report pain compared to children undergoing resin infiltration (RR = 0.21; 95% CI 0.09–0.49; *p* < 0.001) [40]. Both studies did not report other associated adverse effects on resin infiltration [27,40].

### 3.7. Comparison between MITs Alone with MITs in Adjunct to NITs or Another MIT Technique

Two studies [33,41] examined patient-reported outcomes between ART and ART in adjunct with non-invasive or other MID treatments. Sales Huamani et al. (2019) [41] compared the anxiety and stress markers of children when receiving ART only with those receiving ART in adjunct with oral health education. Despite no significant difference in perceived anxiety and cooperativeness between groups [41], the former group had a significantly elevated peri-operative heart rate (*p* = 0.018) and post-operative salivary alpha-amylase levels (*p* = 0.004) [41].

de Souza et al. (2021) compared the time and patient-reported outcomes of children receiving ART with or without Brix3000™ papain gel [33]. No significant difference was found between the pain experience and acceptability with or without Brix3000™ papain gel in adjunct to ART, but the mean treatment time lengthened significantly by 4 min with the application of Brix3000™ papain gel. No adverse effects associated with Brix3000™ papain gel was reported.

## 4. Discussion

This systematic review has inherent strengths and limitations that should be addressed. The protocol and methodology of this review were based on recommendations outlined in the Cochrane Handbook for Systematic Reviews [22]. Furthermore, with independent screening conducted by two reviewers, potential bias in the process of conducting this review could be reduced. Another strength of this review is the use of the Cochrane risk of bias tools for randomized trials (RoB 2.0). Hence, bias could be evaluated under each domain in a holistic manner. The meta-analysis is another strength of this review, which pools data from two studies to evaluate the difference in patient-reported outcomes between MID and conventional treatments.

One limitation of this review is the exclusion of non-English studies, as a few studies were omitted since no English transcripts were found. However, it has been reported that the inclusion of non-English studies may have negligible effect on assessing and summarizing the treatment effect estimates [51]. Furthermore, the overall risk of bias for the included trials are identified as high or with some concerns due to lack of blinding and lack of reporting relevant data. Seven of the studies [37,42,43,45,46,47,48] used observation-based assessment scales to determine the patient’s anxiety which poses a potential increase in the risk of bias. Blinding is difficult as assessors observing the procedure will be aware of the intervention which influences their evaluation of the child’s level of anxiety. Furthermore, these scales could only describe observable signs of dental anxiety and do not take into account the child’s feelings [19]. Observation-based scales also require training and calibration to avoid bias, especially when more than one assessor is involved [19]. The remaining studies utilized self-report assessment scales such as WBFS and FIS and Venham Picture Test. Such scales have the advantage of measuring the cognitive component of fear directly from the patient’s perspective, yet the reliability of reporting from the child might be varied due to maturity and age [19]. Recognizing the benefits and drawbacks of each assessment scale is important to judge the appropriateness of its use in the studies. Hence, the fact that several studies used observation-based scales render a high risk of bias. Ultimately, observation-based scales utilized the patient’s external signs to predict and compare their level of anxiety and pain.

Among all comparisons between MITs, NITs and conventional treatments, it seems relatively obvious that treatments with the least invasive nature are the most likely to render the lowest level of pain and anxiety. It should also be addressed that other confounding factors may have influenced the patient’s perception of anxiety and pain. Other factors, for instance parenting styles, may also influence dental anxiety and behavior, especially for preschool children with no prior dental experience [52]. Factors such as family structure (nuclear or single-parent family) and presence of siblings are determinants of dental fear and anxiety [53]. All of the studies did not delve into such factors and hence such confounders could not be analyzed in this review. Furthermore, varying ages of the patients in the studies of this review is a factor that may affect the patient-reported outcomes and dental anxiety. The current literature has suggested that age influences the patient’s dental anxiety as illustrated by the review conducted by Klingberg and Broberg (2007); they found that the included studies provided evidence to support decreasing dental fear and anxiety with increasing age [54]. In addition, Yon et al. (2020) found that children at three years old displayed more dental anxiety compared to children of older ages [55]. Given the vast age range of participants from 3–10 years old, studies that include older children may have an inherent advantage of more cooperative children who may display less dental anxiety compared to their younger peers.

Moreover, the included studies in this review were graded with high bias or presenting with concern according to the Cochrane risk of bias assessment tool. One large factor is bias in assessing the main outcomes. The previously mentioned seven studies [37,42,43,45,46,47,48] utilizing observation-based scales do not allow adequate blinding of the interventions. One could argue that ultimately, the children could not be entirely blinded from the intervention as they could observe marked differences from various treatment items. For example, multiple studies [25,26,29,32,36,42,43,44,47] included the use of local anesthesia for CR when needed. It has been widely reported that dental anxiety and pain during local anesthesia injection are strongly associated and prevalent [56]. The use of injections during the procedure possibly affects the patient’s perception of the treatment item and greater anxiety and pain levels could be expected. In addition to matters related to blinding and primary outcome reporting, seven studies [29,37,42,43,44,46,47] display concerns in randomization as the process is vaguely outlined. Such detrimental flaws associated with the high risk of bias of the included studies, including but not limited to the non-randomized use of local anesthesia and rubber dam, and the non-standardized treatment locations, should not be underestimated. Thus, the conclusions drawn from these studies were rather weak, which do underline the need of strong research on this issue.

The efficacy of MID to control and arrest caries advancement has been widely reported in the current literature. Published systematic reviews have identified the high efficacy of fissure sealants to prevent or arrest progression of caries lesions by 11–51% compared to no sealant [8,57]. Furthermore, a Cochrane review concluded that micro-invasive treatment modalities such as sealants and resin infiltration are more efficacious than fluoride application in treating initial proximal lesions [18]. Other facets of MID such as HT and selective caries removal are also reported to be highly effective. A recently published systematic review found that lower numbers of failures were associated with HT and selective caries removal compared to CR in primary dentition [58]. Regardless of MID’s PROs and tendency to cause dental anxiety, such highly effective methods should be included in clinical practice.

MITs when used in adjunct with NITs demonstrated significant synergistic effect, as Salas Huamani et al. (2019) reported that the mean heart rate and alpha-amylase levels reduced significantly when ART is used in combination with oral health education [41]. On the other hand, when ART was used together with Brix3000™ papain gel, increased treatment duration but no difference in anxiety level was noted. However, other treatment factors including clinical success and efficacies, cost effectiveness should also be taken into account when clinicians utilized one or more MID modalities to manage the dental decay.

MITs are viable alternatives to manage dental caries in anxious and pre-cooperative children. Clinicians may consider using treatments such as fissure sealant, HT, resin infiltration and NaFV when it is clinically indicated, as it does not cause significantly more fear, anxiety or pain compared to CR. However, relative merits of one MIT over the other could not be observed due to significant inconsistencies found in the included studies. The current findings might only suggest that HT could induce more cooperative behaviors when compared to CR, less pain when compared to ART, and less self-reported anxiety when compared to CSSC. However, the current evidence did not suggest superiority between ART versus CR, and FS versus CR. Nonetheless, all results need to be considered very carefully as they were derived from small-scale studies with high risk of bias and heterogenous study designs.

When delivering dental treatments to pediatric patients, clinicians should first get to know the environmental and personal context that the patient and their families are presented with. Based on the families’ treatment expectations, dental care providers shall then prioritize the treatment goals and decide the kind of care that should be provided. If the main treatment goals are to minimize dental anxiety and enhance cooperation, a detailed dialectic with all possible treatment options, including MITs, general anesthesia and other behavioral management alternatives should be provided to the family before formulating the treatment plan.

Other than the peri-operative patient-reported and observer-based outcomes, the evaluation of patient appreciation and long-term outcomes on both dentist and patient levels should also be taken into account. However, the meta-environmental variables that exist increase the complexity and difficulty in conducting a thorough review of this topic.

Future studies may consider to specifically explore and compare the PROs and children’s anxiety levels of different MID techniques as the range of treatment included in the review is rather broad. Improvement in future study designs and administration, including proper randomization and standardization of the treatment locations, the use of pharmacological and non-pharmacological behavioral management techniques are deemed necessary when investigating dental anxiety, and patient-reported and observer-based outcomes.

## 5. Conclusions

Dental caries is a multi-factorial disease that affects a majority of the pediatric population. The dilemma that pediatric dentists often face is to manage both the disease and the patient in a holistic manner. This review has shown that minimal intervention dentistry is an alternative method to manage dental caries and is not significantly different compared to traditional and non-invasive treatment modalities in terms of patient-reported and observation-based outcomes of pain, anxiety and treatment duration. Given the high efficacy and similar success rates of minimal intervention dentistry compared to other caries management techniques, clinicians may consider employing minimal intervention dentistry techniques in controlling dental caries when the pediatric patients are reluctant to cooperate with traditional approaches to restoring dental caries. However, the findings should be interpreted with cautions as all the included studies had moderate to high risk of bias and the certainties of evidence were low. Well-designed clinical trials are warranted to further investigate the patient-reported and observer-reported outcomes on minimal intervention dentistry.

## Figures and Tables

**Figure 1 healthcare-11-02241-f001:**
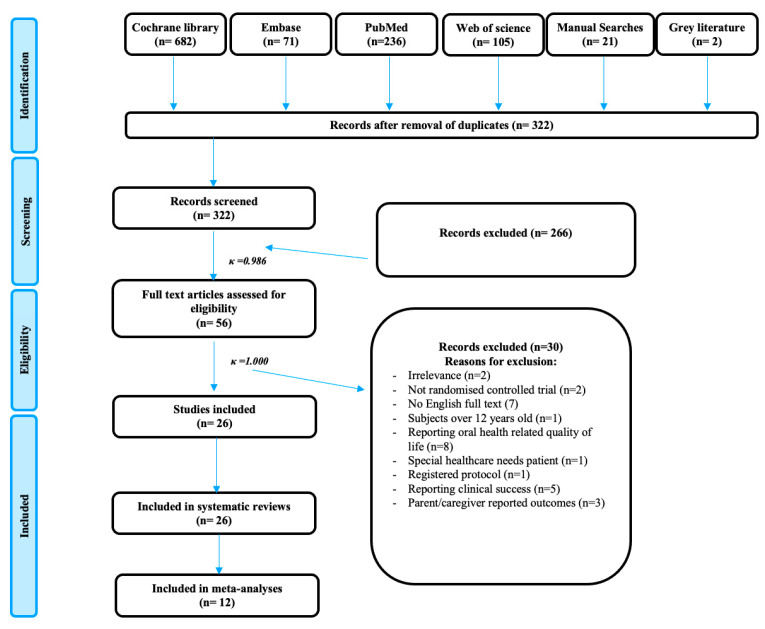
PRISMA flowchart of the systematic review and meta-analysis.

**Figure 2 healthcare-11-02241-f002:**
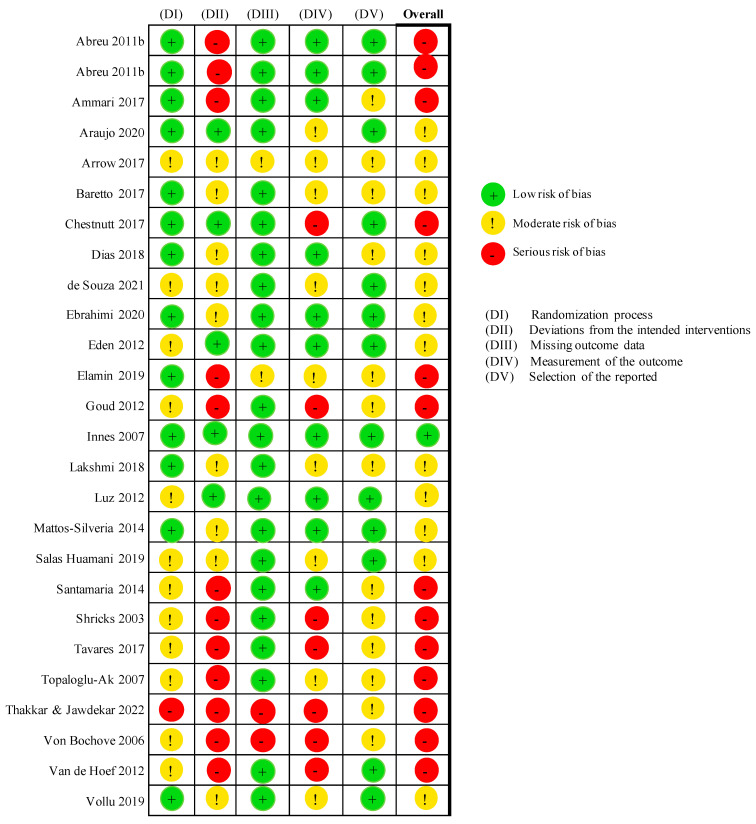
Risk of bias assessment with Cochrane risk of bias assessment tool [14,25,26,27,28,29,30,31,32,33,34,35,36,37,38,39,40,41,42,43,44,45,46,47,48,49].

**Figure 3 healthcare-11-02241-f003:**
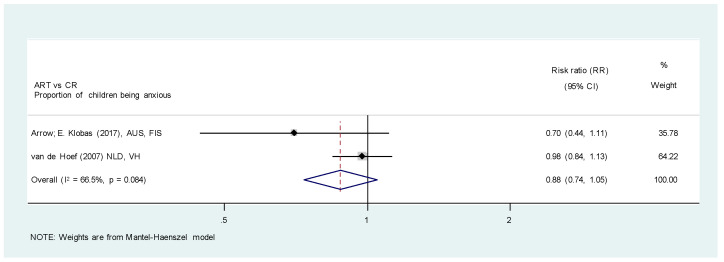
Meta-analysis. ART versus CR. Proportion of children with dental anxiety [29,48].

**Figure 4 healthcare-11-02241-f004:**
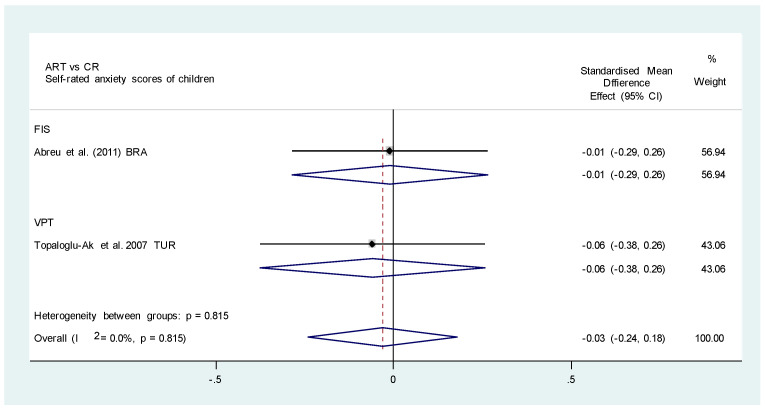
Meta-analysis. ART versus CR. Self-rated anxiety scores of children [25,46].

**Figure 5 healthcare-11-02241-f005:**
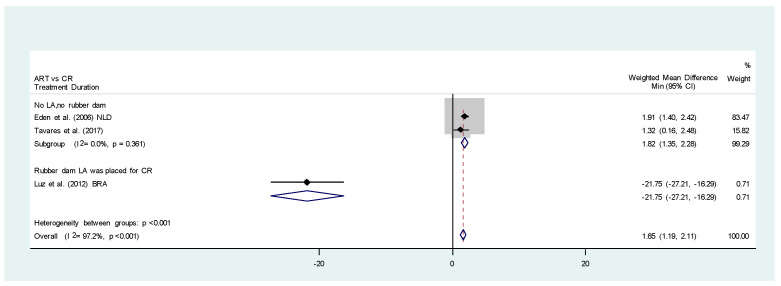
Meta-analysis. ART versus CR. Treatment duration [36,39,44].

**Figure 6 healthcare-11-02241-f006:**
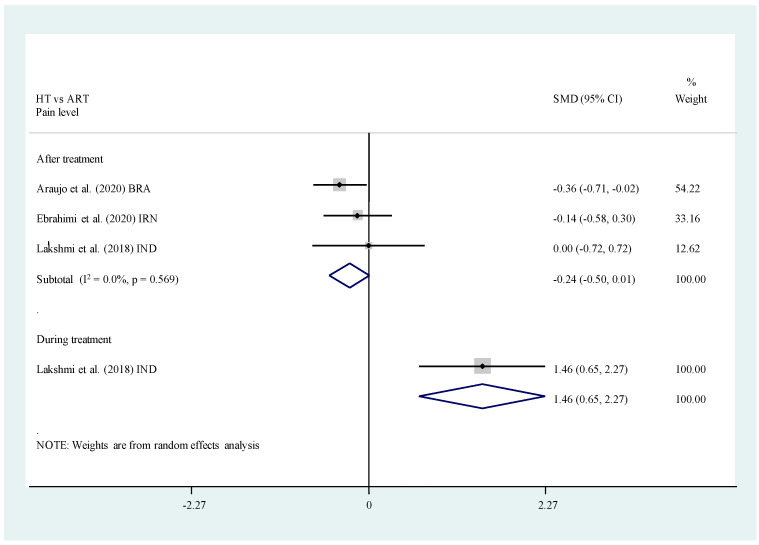
Meta-analysis. HT vs. ART. Self-rated pain level [28,34,38].

**Table 1 healthcare-11-02241-t001:** Characteristics of included studies.

No.	Study (Year and Country)	Number of Participants	N Participants (% M); Range (Mean)	RCT Design, Settings	Method of Assessment	Intervention Group/Control Groups
1	Abreu et al. (2011) Brazil [25]	302	302 (55%); 6–7 years old (6.8)	3 parallel groups, dental clinic	Facial Image Scale (FIS)	(1) Conventional restorative treatment (2) Atraumatic restorative technique (3) Non-invasive treatment (cavity modification)
2	Abreu et al. (2011) Brazil [26]	244	244 (57%) 6–7 years old (6.8)	3 parallel groups, dental clinic	(1) Wong Baker FACES Pain Rating Scale (WBFR)	(1) Conventional restorative treatment (2) Atraumatic restorative technique (3) Non-invasive treatment (cavity modification)
3	Ammari et al.(2017) Germany [27]	50	50 (44%); 5–9 years old (6.2)	Split mouth design, university dental clinic	FIS	(1) Caries infiltration with flossing and fluoride TP (2) Non-invasive flossing and fluoride TP
4	Araujo et al. (2020) Brazil [28]	131	131 (N/A); 5–10 years old (8.1)	2 parallel groups (randomized), university dental clinic	WBFR	(1) ART restoration (2) Hall’s Technique SSC placement
5	Arrow; E. Klobas (2017) Australia [29]	254	254 (59%); 3.7–3.9 (3.8)	Parallel group, dental clinic	FIS	(1) ART (no LA, +/− rotary instruments up to clinician’s judgement (2) Conventional treatment with LA
6	Barretto et al. (2017) Brazil [30]	94	94 (51); 6–8 years old (no mean provided)	Analytical cross sectional study, randomized	FIS	(1) ART restoration (2) Non-invasive SDF application
7	Chestnutt et al. (2017) United Kingdom [31]	1015	1015 (46.5); 6–7 years old (no mean provided)	Parallel group, NHS Mobile dental unit at schools	(1) Delighted Terrible Faces Scale (DTF Scale)	(1) Non-invasive fluoride varnish (2) Resin Pit and Fissure Sealant
8	Dias et al. (2017) Brazil [32]	28	28 (53.6); 3–8 years old (6.79)	Parallel group, university dental clinic	FIS	(1) Sealing with flowable resin of occlusal surface (2) Resin composite restoration with sealing of flowable resin
9	de Souza et al. (2021) Brazil [33]	20	20 (60) 3–9 years old	2 parallel groups, pediatric dental clinic	(1) Time (2) FLACC-r score (3) Hedonic Facial Scale	(1) ART + Brix3000™ group (2) ART-only group
10	Elamin et al. (2019) Sudan [34]	164	164 (50.4), 5–8 years old (NR)	Parallel groups, general dental clinic	FIS	(1) Hall’s Technique (HT) (2) Conventional SSC placement (CT)
11	Ebrahimi et al. (2020) Iran [35]	123	123 (37.4), 4–9 years old	3 parallel groups, university dental clinic	(1) FACES Pain Scale revised (2) Time	(1) Hall’s Technique (HT) (2) Conventional SSC placement (CT) (3) GIS restoration
12	Eden et al. (2006) NLD [36]	157	157 (48%) 7 years old (7.0)	Split mouth design with washout, university dental clinic	(1) Time	(1) ART (2) conventional restoration without LA
13	Goud et al. (2012) India [37]	200	200 (no data); 6–8 years old (no mean provided)	Parallel group, hospital dental clinic	(1) Venham Scale	(1) ART restoration (2) Non-invasive treatment (cavity modification)
14	Innes et al. (2007), UK [14]	257	132 (52.2) 3–10 years (6.8)	Split mouth, general dental practice	(1) Time	(1) Hall’s Technique (2) Conventional SSC
15	Lakshmi et al. (2018) India [38]	30	30 (NR); 5–8 years old (NR)	2 parallel groups, clinic study	WBFR	(1) ART restoration (2) Hall’s Technique SSC
16	Luz et al. (2012) Brazil [39]	30	30 (43.3), 4–7 years old (NR)	2 parallel groups, clinic study	Time	(1) ART (2) Conventional restoration
17	Mattos-Silveria et al. (2014) BRA [40]	141	141 (47.5%): 3–10 years old (6.56)	3 parallel groups, university dental clinic	WBFR	(1) Non-invasive flossing instruction (2) Non-invasive SDF application (3) Resin infiltration
18	Salas Huamani et al. (2019) Brazil [41]	78	78 (48.8) 6–8 years old (6.5)	2 parallel groups, school	(1) Modified Venham Picture Test (2) Modified Venham Anxiety Scale (3) Heart Rate (HR) (4) Salivary Cortisol and Alpha-Amylase Levels	(1) OHES + ART-group (2) ART-group
19	Santamaria et al. (2014) Germany [42]	169	169 (56.8%); 3–8 years old (5.55)	3 parallel groups, university dental clinic	(1) Frankl Scale (2) Visual Analogue Scale of Faces	(1) Conventional restorations (2) Hall’s Technique SSC (3) Non-invasive treatment (cavity modification)
20	Shricks et al. (2003) Indonesia [43]	403	403 (51.6%); 4–7 years old (6.3)	2 parallel groups, hospital dental clinic	(1) Venham index	(1) Conventional restorations (2) ART restoration
21	Tavares et al. (2017) Brazil [44]	79	79 (36.7%); 5–8 years old (6.6)	Split mouth design with washout, university dental clinic	(1) FIS (2) WBFR	(1) Conventional restorations (2) ART restoration
22	Thakkar and Jawdekar (2022) India [45]	60	60 (48%), 7–8 years old (7.6)	2 parallel groups, university dental clinic	(1) Frankl Scale (2) Time	(1) HT (2) Conventional restorations
23	Topaloglu-Ak et al. (2007) Turkey [46]	160	160 (N/A) 6–7 years old	2 parallel groups, university dental clinic	(1) Venham Picture Test	(1) Conventional restorations (2) ART restorations
24	Van Bochove et al. (2006) Netherlands [47]	300	300 (48%); 6–7 years old (6.98)	4 parallel groups, university dental clinic	(1) Venham index (2) Venham Picture Test	(1) Conventional restorations with LA (2) Conventional restorations without LA (3) ART with LA (4) ART without LA
25	van der Hoef (2007), NLD [48]	299	299 (51.8) 6–7 years (7.5)	4 parallel groups, university dental clinic	Venham index	(1) Conventional restorations with LA (2) Conventional restorations without LA (3) ART with LA (4) ART without LA
26	Vollu et al. (2019) Brazil [49]	68	68 (61.2); 2–5 (no mean provided)	Parallel group, university clinic	FIS	(1) Non-invasive SDF application (2) ART restoration

**Table 2 healthcare-11-02241-t002:** GRADE summary of findings table.

Comparison	N Study	N Teeth	Outcome	Mean Score of Intervention	Risk of Bias ^a^	InconsistencyHeterogenicity ^b^	Indirectness ^c^	Imprecision ^d^	Publication Bias ^e^	Quality of Evidence(GRADE)
I^2^ (%)	χ ^2^ Test (*p* Value)
ART vs. CRProportion of children being anxious	2	425	No difference	RR:0.88 (0.74, 1.05)	Serious 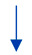	66.5%Serious 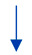	0.084--	Serious 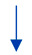	Not serious--	N/A	⊕OOO very low due to overall high risk of bias, inconsistency, imprecision and indirectness
ART vs. CRSelf-rated anxiety levels of children	2	371	No difference	SMD−0.03 (−0.24, 0.18)	Serious 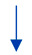	0.0%--	0.815--	Not serious--	Serious ^#^ 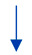	N/A	⊕⊕OO low due to overall high risk of bias and imprecision
ART vs. CR (without LA and RD)Treatment duration	2	516	CR required significantly shorter treatment time	WMD 1.82 (1.35, 2.28)	Serious 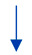	0.0%	0.361--	Not serious--	Not serious--	N/A	⊕⊕⊕O moderate due to overall high risk of bias
HT vs. CSSCTreatment duration	2	381	No difference	WMD 1.71 (1.21, 2.21)	Serious 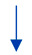	97.9% * 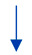	<0.001 **	Not serious--	Serious ^#^ 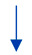	NA	⊕OOO very low due to overall high risk of bias, inconsistency, and imprecision
ART vs. CRSelf-rated pain levels of children	3	240	No difference	SMD 1.34(−0.50, 0.01)	Serious 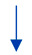	0.0%	0.569	Not serious--	Serious ^#^ 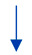		⊕⊕OO low due to overall high risk of bias and imprecision

^a^ Considered as serious if overall half of the studies included were of serious risk of overall bias. ^b^ Inconsistency: considered as serious when I^2^ statistics > 60% (*) and *p*-value of χ^2^ test < 0.05 (**). ^c^ Indirectness: considered as serious when applicability of findings was restricted in terms of population, intervention, comparator and outcomes. ^d^ Imprecision: considered as serious when total number of events was below 300 for dichotomous outcomes or 400 for continuous outcomes (#), or when the upper and lower limits of 95% CI include both meaningful benefits and harm. ^e^ Publications bias: considered as serious if *p*-value of Begg’s funnel plot < 0.05. Not applicable (N/A) if funnel plot could not be constricted given the limited numbers of study. Publication bias was difficult to detect and thus no downgrading was performed. ↓: downgrade by one level in quality of evidence. --: no change in quality of evidence.

## Data Availability

The data presented in this study are available in Appendix A.

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
