# Peer review of "The Impact of Minimal Intervention Dentistry on Patient-Reported and Observation-Based Outcomes in the Pediatric Population: A Systematic Review and Meta-Analysis"

_healthcare, 2023, doi:10.3390/healthcare11162241_

Round 1

Reviewer 1 Report

It is obvious that the authors have done a lot of work in evaluating the studies for this review. They have clearly stated the many inherent biases not the least of which is the many studies completed in dental schools or university settings.  Evaluating behavior is hard enough let alone with student providers who themselves are likely very anxious. Thus, not surprising that no statistically significant differences were found in most of the studies. Given the findings, the only addition I would suggest the authors provide is a more robust explanation of the pathway for care. Any decision on the kind of care that is provided must be based on the environmental and personal context that a person / family unit presents with.  Long term goals and outcomes taken with the context of these domains. If decrease in anxiety and increase in co-operation a goal, then MIT is an option within the full spectrum of the dialectic that should be given. This would include general anesthesia predicated again on long term outcomes. The meta environmental variables that exist across many of the studies speaks to the complexity of this evaluation and as such to have an impact on the practising clinician perhaps deserves more explanation or discussion.   

Reviewer 3 Report

Clear focused question is not mentioned. If authors have done multiple comparison what was the result of different invasive approach try to add on more. Figure 1 is not proper. overall, this systematic review has followed the PRISMA guidelines and all the steps and results are mentioned clear and concisely. 

minor editing is required. 
